# Comparative Analysis of Floral Transcriptomes in *Gossypium hirsutum* (Malvaceae)

**DOI:** 10.3390/plants14040502

**Published:** 2025-02-07

**Authors:** Alexander Nobles, Jonathan F. Wendel, Mi-Jeong Yoo

**Affiliations:** 1Chemistry & Biomolecular Science Department, Clarkson University, Potsdam, NY 13699, USA; noblesa@upstate.edu; 2Department of Ecology, Evolution, and Organismal Biology, Iowa State University, Ames, IA 50011, USA; jfw@iastate.edu; 3Biology Department, Clarkson University, Potsdam, NY 13699, USA

**Keywords:** MADS-box genes, *Gossypium hirsutum*, flower development, polyploidy, homoeologs

## Abstract

Organ-specific transcriptomes provide valuable insight into the genes involved in organ identity and developmental control. This study investigated transcriptomes of floral organs and subtending bracts in wild and domesticated *Gossypium hirsutum*, focusing on MADS-box genes critical for floral development. The expression profiles of A, B, C, D, and E class genes were analyzed, confirming their roles in floral organ differentiation. Hierarchical clustering revealed similar expression patterns between bracts and sepals, as well as between petals and stamens, while carpels clustered with developing cotton fibers, reflecting their shared characteristics. Beyond MADS-box genes, other transcription factors were analyzed to explore the genetic basis of floral development. While wild and domesticated cotton showed similar expression patterns for key genes, domesticated cotton exhibited significantly higher expression in carpels compared to wild cotton, which aligns with the increased number of ovules in the carpels of domesticated cotton. Functional enrichment analysis highlighted organ-specific roles: genes upregulated in bracts were enriched for photosynthesis-related GO terms, while diverse functions were enriched in floral organs, supporting their respective functions. Notably, A class genes were not significantly expressed in petals, deviating from the ABCDE model, which warrants further analysis. Lastly, the ABCDE class genes exhibited differential homoeolog expression bias toward each subgenome between two accessions, suggesting that the domestication process has influenced homoeolog utilization despite functional constraints in floral organogenesis.

## 1. Introduction

Floral development is a critical process for reproduction in angiosperms. It is a complex process that is controlled and regulated by the expression of networks of genes and regulatory controls including transcription factors. The transcription factors that are responsible for floral organ specification are usually from highly conserved gene families. MADS-box genes are key players in floral development regulation as well as other developmental processes in plants [1,2,3]. MADS-box genes contain a conserved sequence domain (M) that is 60 amino acids long, which remains nearly unchanged throughout the course of evolution due to its importance in growth and development in eukaryotes. There are two monophyletic lineages in the MADS-box gene family, type I and II [4,5]. In plants, the MIKC-type genes, one of two clades of type II MADS-box genes [6,7], are well studied due to their significant role in floral transition and flower formation. They are characterized by three additional domains other than M domain, the intervening (I), keratin (K), and C-terminal (C) domains [1,8,9]. Among these domains, the K domain is important for protein dimerization, while the C-domain mediates protein–protein interactions [10,11,12,13].

Floral transition is mediated by several MIKC-type MADS-box genes, for example, FLOWERING LOCUS C (FLC), SHORT VEGETATIVE PHASE (SVP), SUPPRESSOR OF OVEREXPRESSION OF CONSTANS 1 (SOC1), and AGAMOUS-LIKE 24 (AGL24) in Arabidopsis thaliana [14,15,16]. Floral organ formation and development were initially explained by the “ABC” model, based on the discovery of mutants [17,18], but this has been replaced by a more generic model, the ABCDE model [7,10,19,20,21]. Based on this model, the combination of the ABCDE class genes determine each floral organ identity as follows: sepals (A + E), petals (A + B + E), stamens (B + C + E), carpels (C + E), and ovules (D + B_sister_ + E) through the formation of unique protein tetramers [17,22,23,24,25]. The action of these tetramers as transcription factors guiding floral development is known as the quartet model. This model explains that each whorl expresses different amounts and combinations of the five types of transcription factors (A, B, C, D, and E class genes) and thus each whorl will have unique transcription factor tetramers and develop into a different floral organ [2,10,19]. For example, stamen identity is determined by a tetramer of two B class genes, APETALA3 (AP3) and PISTILLATA (PI), one C class gene, AGAMOUS (AG), and one E class gene, SEPALLATA (SEP), while carpel identity is specified by a tetramer consisting of AG and SEP proteins [19]. However, MADS-box genes are not the only transcription factors involved in floral development; for example, BELLRINGER (BLR), JAGGED (JAG), ETTIN (ETT), and Repressor of Gibberellic Acid (RGA) are also known. These transcription factors can contribute to flower and inflorescence meristems (BLR) [26,27,28] or control inflorescence development (RGA) [29,30].

Upland cotton, *Gossypium hirsutum* L., is an allopolyploid species containing two subgenomes, an A genome and a D genome, reflecting the origin of the polyploid lineage from hybridization and genome doubling from maternal (A) and paternal (D) diploid parents [31,32,33,34]. These genomes are derived from a polyploidization event approximately 1–2 million years ago, when a cotton species resembling *G. raimondii* (D_5_), which has the D genome, hybridized with a species resembling *G. herbaceum* (A_1_) and *G. arboreum* (A_2_), which carry the A genome [31,32,33,34]. This event led to the speciation of at least six different species of cotton with *G. hirsutum* as the most widely cultivated species [33,34,35,36]. Seed fibers, the most important raw material for the global cotton industry, are single-celled trichomes growing from ovular epidermal cells, of which fiber development is controlled by MADS-box genes. Notably, the development of seed and fibers is closely intertwined with carpel development. Consequently, research has primarily focused on MADS-box genes involved in carpel and fiber development, such as the *AG* homolog (*GhMADS3*) [37], *SHATTERPROOF* (*SHP*; *GhMADS7*) and *SEEDSTICK* (*STK*; *GhMADS5*, *GhMADS6*) homologs, which exhibit high expression in carpel and developing fibers [38,39]. Additionally, the *AP3* homolog (*GhMADS9*) [40], which is related to petal and stamen organ identify, has been studied. Jiang et al. [41] identified and analyzed the expression of 53 MIKC-type genes across various organs, including flowers, individual floral organs, fibers, and ovules, revealing expression patterns broadly consistent with those observed in *Arabidopsis*. Two genome-wide analyses of MADS-box genes have been performed in *Gossypium*, which identified 106 to 110 MIKC-type MADS-box genes belonging to 13 subfamilies [42,43]. Both studies investigated transcriptome profiles using an RNA sequencing (RNA-Seq) technique, either focusing on floral buds or various tissues including flowers. They confirmed that most MIKC-type MADS-box genes of *G. hirsutum* were generally expressed in flower, ovule, and fiber tissues [43] or in the developing floral buds [42]. They also showed that the gene expression patterns of several MIKC-type MADS-box genes were consistent with the ABCDE model in floral organogenesis. However, to date, the expression levels of the duplicated ABCDE genes (A and D homoeologs) in allopolyploid cotton have not been studied at the genome-wide level.

Here, we aimed to analyze transcriptome profiles of each floral organ, including sepals, petals, stamens, and carpels, in both wild and domesticated *G. hirsutum* using RNA-Seq. The wild form of *G. hirsutum* accession TX2094 is distinguished by yellow anthers and prominent red petal spots at the base of its flower (Figure 1A). In contrast, the flower of domesticated *G. hirsutum* is characterized by creamy anthers and the absence of petal spots (Figure 1B). These differences highlight evolutionary adaptations and domestication processes that may have influenced floral gene expression and regulation.

To investigate these changes, we evaluated the ABCDE model using genome-wide data and compared the expression levels of both homoeologs of ABCDE class genes across floral organs. We also explored other transcription factors that might be involved in floral organogenesis through functional enrichment analyses of transcripts differentially expressed in each floral organ relative to bracts. Characterizing the various transcription factors will facilitate an understanding of polyploidy and the molecular processes that drive floral development and tissue differentiation.

## 2. Results

### 2.1. Overall Transcript Expression

A total of 76,018,537 clean reads were obtained from bracts and floral organs, and on average, 72.48% of these reads were successfully mapped onto the reference genome of *G. hirsutum*. In all organs, more reads were mapped onto the D subgenome compared to the A subgenome (Appendix A), as shown previously [44]. Among transcripts with Transcripts Per Kilobase Million (TPM) ≥ 2 [45], about 22,485 (29.9%)–34,543 (45.8%) were expressed in the five organs; the highest number was retrieved from carpels, which includes stigma, style, ovary, and ovule, while petals and stamens had 22,760 and 22,485 transcripts expressed in TX2094 (Appendix A). However, in TM-1, the highest number was observed in sepals and the lowest numbers in stamens (Appendix A). The difference might come from sample differences between two accessions (Figure 1). For example, the carpels of TM-1 contained ovules only on the day of flowering, also referred to as 0 Days Post-Anthesis (DPA), while carpels of TX2094 consisted of stigma, style, ovary, and ovules (Figure 1), which explains a higher transcriptome complexity compared to that of TM-1 (Appendix A).

When we considered expressed transcripts (≥2 TPM) with 22,384 homoeologous gene pairs, all organs in both accessions expressed more genes derived from D subgenome (Figure 2), as shown previously [44]. However, these biases were significant in bracts and sepals of TX2094 and all the organs investigated in TM-1 (Figure 2).

Transcript expressions were compared across the five organs to study differential expressions. In TX2094, bracts and sepals were most similar to each other, as were stamens and petals. For example, bracts and sepals have 6398 differentially expressed transcripts (8.5% of 75,376 transcripts) with 2935 upregulated in sepals and 3463 upregulated in bracts (Appendix A). The greatest differentiation was observed between carpels and petals, which showed 18,383 differentially expressed transcripts (24.4% of 75,376 transcripts; Appendix A). In TM-1, similar results were retrieved for carpels and petals, but stamens and petals showed the least differentiation, followed by bracts vs. sepals (Appendix A). In agreement with these results, bract and sepal, and stamen and petal, clustered together based on overall transcriptome profiles in both TX2094 and TM-1 (Appendix A).

We further examined differentially expressed transcripts in each floral organ using bracts as a common reference (Appendix A). Results showed that there were a higher number of upregulated transcripts in bracts compared to floral organs than vice versa (Appendix A). There were 1960 and 2577 transcripts upregulated in bracts of TX2094 and TM-1, respectively (Appendix A), while only 398 and 606 genes were downregulated in bracts of TX2094 and TM-1 compared to floral organs (Appendix A). That is, only 398 genes in TX2094 and 606 genes in TM-1 were upregulated across four floral organs when compared to bract transcriptomes. TX2094 and TM-1 showed differential expressions as expected from their morphological differentiation, such as yellow vs. creamy anthers and presence and absence of petal spots. For example, although they commonly had 2371 transcripts upregulated in carpels compared to bracts, each accession had 3559 and 6213 uniquely differentially expressed transcripts in TX2094 and TM-1, respectively (Appendix A).

### 2.2. Functional Analysis of Differentially Expressed Transcripts

Functional enrichment tests were performed for differentially expressed transcripts in each floral organ relative to bracts using agriGO v2.0. Results showed that many transcripts upregulated in bracts when compared to floral organs are involved in photosynthesis, while transcripts upregulated in floral tissues have a wide variety of functions and purposes in both accessions (Appendix A). Many transcripts upregulated in floral organs compared to bracts are involved in floral organ development, such as microsporogenesis, plant-type spore development, carpel development, and floral organ formation in both accessions (Appendix A). We further investigated differentially expressed transcripts using STRING v.12.0. The biological process (BP) enrichment tests showed that different GO terms were enriched for floral organs in TX2094 and TM-1, except in sepals and carpels. In sepals, most GO terms are related to “response to chemical” and “response to abiotic stress” in both accessions, while in carpels “cell division” and “cell cycle” were enriched in both accessions (Appendix A). However, “phenylpropanoid metabolic process” was additionally enriched in petals of TX2094, while “cell tip growth” and “pollen tube growth” were over-represented in stamens of TX2094 (Appendix A). In TM-1, “organic acid catabolic process” and “small molecule catabolic process” were enriched in both petals and stamens (Appendix A). Similar to the results of agriGO, transcripts upregulated in bracts compared to floral tissues showed many related to photosynthesis in both accessions (Appendix A). Transcripts upregulated in sepals relative to bracts were enriched for “response to chemical” and “response to abiotic stimulus” in both accessions (Appendix A) as well as “phenylpropanoid biosynthesis” (Appendix A). In both TX2094 and TM1, petal transcripts were involved in similar biological processes enriched in sepals (Appendix A) and “flavonoid synthesis” (Appendix A). Transcripts from stamens were enriched for “pollen tube growth” in TX2094 and “response to abiotic stress” in TM-1 (Appendix A). In carpels, enriched terms were mostly related to “cell division and microtubule motor activity” in both accession (Appendix A) and additionally “flavonoid biosynthesis” in TX2094 (Appendix A).

### 2.3. Differentially Enriched Transcription Factors

As transcription factors play important roles in plant growth and development, we tested whether specific transcription factors were enriched in each organ. There were several transcription factors enriched in upregulated transcripts in bracts compared to floral organs, such as CONSTANS (CO)-like, three-amino-acid-loop-extension (TALE), NY-FA, TCP, and WRKY in TX2094 (Figure 3A; Appendix A). These transcription factors are diverse in function but include those involved in controlling meristem identity and formation, organ morphogenesis, stress response, flowering time regulation, and plant development form [46,47,48]. In TM-1, CO-like, TCP, and WRKY, but not NY-FA and TALE, were enriched in bracts, but additionally ethylene response factors (ERFs) were also enriched (Figure 3B, Appendix A). ERFs play important roles in plant growth, development, and environmental stress responses [49,50]. In contrast, MIKC-type MADS-box gene transcription factors were enriched in all floral organs compared to bracts in both accessions (Figure 3A,B), but significance levels were much lower in TM-1 compared to TX2094 (Appendix A). NAM, ATAF, and CUC (NAC) transcription factors were also enriched in all floral organs except carpels relative to bracts in TX2094, while NAC transcription factors were most significantly enriched in carpels in TM-1 (Appendix A). NAC transcription factors regulate plant development, morphogenesis, and abiotic and biotic stress tolerances [51,52,53,54]. Interestingly, NAC transcription factors were commonly over-represented in petals, stamens, and carpels compared to other floral tissues, but not in sepals (Appendix A). In sepals, transcription factors similar to those enriched in bracts were over-represented, as supporting their similar transcriptomic profiles. In carpels, several distinctive transcription factors were enriched, including auxin response factors (ARF), BARLEY B RECOMBINANT/BASIC PENTACYSTEINE (BBR-BPC), growth regulating factors (GRF), and ZF-HD (Figure 3C,D, Appendix A), but ERF and YABBY were under- and over-represented, respectively, only in TX2094 (Figure 3C, Appendix A). Further results for selected transcription factors are described below.

To further investigate the functional relationships among differentially expressed transcription factors, we performed STRING analysis. This analysis revealed diverse pathways involved in each floral organ’s development. Notably, MADS-box genes exhibited the strongest interactions, and a major cluster emerged comprising most transcription factors implicated in flowering and floral organogenesis, which interconnected with clusters of various other transcription factor types (Appendix A).

#### 2.3.1. MIKC-Type MADS-Box Gene Expression in Floral Organs

The differential expressions of transcription factors, particularly MIKC-type MADS-box genes, were a primary focus. First, we confirmed the members of MIKC-type MADS-box genes in *G. hirsutum* by performing phylogenetic analysis (Appendix A). Compared to a previous analysis [42], we identified two more *SEP* genes (12 vs. 10 genes), one more *AGL6*-like gene (8 vs. 7 genes), and one less *AP1/FUL*-like gene (10 vs. 11 genes).

Hierarchical clustering analysis of MIKC-type MADS-box gene expression profiles showed that bracts and sepals were similar in their gene expression profiles, while petals and stamens shared similar gene expression (Appendix A), in agreement with the entire transcriptomes (Appendix A). Carpels, containing ovules with fiber cells on their surface, clustered with developing cotton fibers. The expected classes of MADS-box genes were mostly expressed in the appropriate tissue types according to the ABCDE model, except A class genes (Appendix A). The MADS-box genes are separated into two primary clades based on their expression patterns. Cluster I is composed of B and E class genes (*SEP1/2/3*) (Appendix A). In cluster II, cluster II-a encompasses C, D, and B_sister_ class genes along with some B and E class genes, while cluster II-b consists of two subclusters; one contains all low-expressed genes, and another one includes A and E class genes (*SEP4*) (Appendix A).

We further examined the expression of each gene class. A class genes consist of *FRUITFUL* (*FUL*)-like and *euAP1*-like gene groups, *FUL* and *AP1*, respectively [55]. There are 12 A class gene homologs in *G. hirsutum*, of which a clade of *Gohir.A07G079100* and *Gohir.D07G084400* was sister to a clade of *FUL*-like and *euAP1*-like clades (Appendix A). We observed that most A class gene homologs of *G. hirsutum* exhibited the highest expression in bracts and sepals, not sepals and petals as predicted by the ABCDE model (Figure 4A,B). Interestingly, the A class gene homologs showed low expression in petals of TX2094 and TM-1 in this study (<15 TPM) (Figure 4). In addition, *AP1.2*, *AP1.3*, and *FUL.2* were expressed in bracts and sepals of both accessions, and additionally in carpels of TX2094 (Figure 4A) and in petals of TM-1 (Figure 4B). However, *FUL.1* was expressed in bracts and sepals of TX2094 (Figure 4A). The previous studies showed that *AP1.3* was expressed in bracts, sepals, and petals [42], while *FUL.2* was detected in sepals, petals, and carpels [43]. Four *AP1* homologs (*AP1.1*, *CAULIFLOWER* (*CAL*)) exhibited much lower expression in both accessions when compared to other A class gene homologs (Figure 4). Their low expression might be due to absence of both *euAP1* and *FUL* motifs at the C-termini required for their proper protein structure [55] (Appendix A). Lastly, the expression levels of A class gene homologs were much lower in TM-1 compared to those of TX2094 (Figure 4).

B class genes consist of two members: *AP3* and *PI*. As expected, B class gene homologs were highly expressed in petals and stamens, except *AP3.3,* which showed relatively low expression in both accessions (Figure 4C,D). Also, *AP3.3* homologs do not have *PI*-derived and *AP3* motifs (Appendix A). In both accessions, the D homoeolog of *AP3.2* showed much higher expression compared to their counterparts (Figure 4C,D). TM-1 showed expression patterns similar to those of TX2094, generally displaying higher expression levels. However, the D homoeolog of *AP3.2* in TM-1 showed much lower expression to TX2094 (Figure 4D).

C class genes, comprising *AG* and *SHP*, were primarily in stamens, carpels, and developing fibers (Figure 5A,B and Appendix A). D class gene homologs, *STK*, and B*_sister_* genes, *TRANSPARENT TESTA16* (*TT16*), were exclusively expressed in carpels and developing fibers (Figure 5A,B and Appendix A). While both homologs of *TT16* were expressed in carpels of TM-1, one homolog was not expressed in TX2094 (Figure 5A). Notably, TM-1 consistently exhibited higher expression levels across all gene homologs compared to TX2094 (Figure 5), aligning with the increased ovule number observed in domesticated cotton.

E class genes consist of four members: *SEP1*, *SEP2*, *SEP3*, and *SEP4*. These genes were expressed in all floral organs, with *SEP4* also showing expression in bracts (Figure 5C,D), which supports its distinct clade within cluster II (Appendix A). The two accessions exhibited similar expression patterns, except for the D homoeolog of *SEP4.1*, which displayed higher expression in TX2094 compared to TM-1 (Figure 5C,D).

The *AGL6* clade, which is sister to the *SEP* clade (Appendix A), contains eight homologs in cotton. In contrast to redundant expressions of *AGL6* and *SEP* genes in other plants, *G. hirsutum AGL6* gene homologs exhibited variable expression patterns, with predominant expression in sepals and additional expression in bract, petals, and carpels (Figure 6A,B). The expression levels of *AGL6* were notably lower than those of *SEP* genes. In both accessions, the D homoeolog of *AGL6.1* showed much lower expression compared to the A homoeolog, possibly due to the truncated sequence at the C-terminus (Appendix A). Expression patterns differed between the two accessions: *AGL6.2* genes exhibited higher expression levels in TX2094 compared to TM-1, while *AGL6.4* homologs showed higher expression across all floral organs in TM-1 compared to TX2094 (Figure 5B).

Two classes of flowering time genes, *SVP* and *SOC*, were included in cluster II-b*i*, but their expression was relatively low (<20 TPM) and mainly expressed in vegetative tissues (leaf and bracts; Appendix A). Other transcription factors known to be involved in floral development, *BLR*, *JAG*, *ETT*, and *RGA*, also exhibited low expression (<10 TPM).

#### 2.3.2. Other Transcription Factors

Alongside MIKC-type MADS-box genes, NAC transcription factors were also enriched and exhibited upregulation in all floral organs, except carpels of TX2094, compared to bracts (Appendix A). The expression patterns of *NAC* transcription factors were analyzed using a heatmap, revealing two distinct clusters. Transcripts in cluster I exhibited higher expression in floral organs, whereas those in cluster II generally showed lower expression levels. However, two transcripts, *Gohir.A12G013900* and *Gohir.D12G014600*, stood out with high expression in the bracts and sepals of TX2094 (Appendix A). While NAC transcription factors are commonly over-represented in petals and stamens compared to other floral organs, their various roles in plant development and responses to abiotic and biotic stress [53,56,57,58] complicate the inference of their specific function in petals and stamens.

Among six transcription factors enriched in carpels, we investigated the expression patterns of BBR-PPC and YABBY transcription factors, as they are known to be involved in ovule development. BBR-PPC proteins have various regulatory roles in plant growth and development, for example, targeting many genes of brassinosteroid signaling [59]. One of seven *Arabidopsis BBR-PPC* genes regulates *STK* [60], which controls ovule identity. Twelve *BBR-PPC* gene homologs were identified, and they were mainly expressed in sepals and carpels of both accessions, with the highest expression levels in carpels (Figure 7). Similar expression patterns were observed in TX2094 and TM-1; however, the expression levels were higher in TM-1 than in TX2094 (Figure 7). Notably, their expression levels peaked just before anthesis or on the day of flowering, then decreased as DPA progressed (Figure 7), suggesting a role in the early stages of ovule development.

In *G. hirsutum*, 24 YABBY transcription factors were identified, indicating gene family expansion in this species. As *G. hirsutum* is a tetraploid species, the presence of duplicate genes is expected. However, except for *INNER NO OUTER* (*INO*) and *CRABS CLAW* (*CRC*), more than two homoeologs were identified in the *YABBY* gene family. Specifically, three *YAB1*, two *YAB2*, and five *YAB5* homoeologous gene pairs were detected (Figure 8A). The expressions of 24 *YABBY* gene homologs were variable across the organs. First, cluster I was divided into two subclusters. Transcripts in cluster I-a were highly expressed in bracts and sepals of both accessions, while transcripts in cluster I-b were mainly expressed in all the organs, except in stamens of both accessions and carpels of TM-1 (Figure 8B). For example, *YAB2-2* homologs were expressed in bracts and sepals of both accessions, and carpels of TX2094, and *YAB2-1* homologs showed the highest expression in carpels of TX2094 among all YABBY transcription factors (Figure 8B). Cluster II formed two main clusters. In cluster II-a, *YAB5-1* homologs were highly expressed in stamens of TM-1, while homologs of *YAB1-1* and *CRC* in cluster II-b were mainly expressed in carpels (Figure 8B) in agreement with the previous study [61]. However, none of YABBY showed high expression in carpels of TM-1. Cluster II-c includes *INO*, *YAB1-2*, and most *YAB5* homologs, of which expression was relatively low compared to other YABBY transcription factors (Figure 8B).

### 2.4. Homoeolog Expression Bias (HEB) in Floral Organs

#### 2.4.1. Overall Comparison of HEB Between TX2094 and TM-1

We also investigated homoeolog expression bias (HEB) of 22,384 homoeologous gene pairs in TX2094 and TM-1. HEB is defined as statistically unequal expression levels of two parental copies (homoeologs) in a polyploid species [62]. The highest HEB was observed in sepals of both accessions, and HEB was much more frequent in TM-1 than TX2094 (Figure 9A,B). However, when HEB was evaluated for 2123 homoeolog gene pairs of transcription factors, we observed more HEB in TX2094 with balanced HEB toward each subgenome (Appendix A). Three organs, including bracts, sepals, and carpels, exhibited HEB toward the D subgenome in both accessions (Figure 9), in agreement with previous studies [44,63,64,65]. However, a lower and higher degree of HEB in some organs was observed, for example, 14.1% and 34.9% of HEB in petals of TX2094 and TM-1, respectively (Figure 9A,B), but 19.8% and 20.6% of HEB was reported in petals of TX2094 and Maxxa, respectively [65].

When comparing HEB between TX2094 and TM-1, 26.5% of HEB was shared in sepals, whereas only 15.1% was common in carpels (Table 1). The lower proportion of shared HEB in carpels may partly be attributed to differences in the materials analyzed. For example, the carpels of TX2094 included stigma, style, ovary, and ovules (Figure 1C), while in TM-1, only ovules at 0 dpa were analyzed [66]. In contrast, 17.7% of HEB was common in stamens, suggesting that stamens and carpels may have undergone greater differentiation than other organs, perhaps because of domestication, as these organs are potentially more associated with domestication traits. In general, as many HEB were unique to each accession—with TM-1 showing a significantly higher number of HEB—the domestication process might have driven extensive changes in HEB (Table 1).

To assess whether specific GO terms were enriched for HEB toward each subgenome, a functional enrichment test was carried out with STRING. Although there were higher number of GO terms enriched in TM-1, both accessions shared similar enriched GO terms in each category. This suggests that similar function pertaining homoeolog pairs were recruited for HEB in TX2094 and TM-1. We further examined HEB in TM-1, for example, comparing enriched GO terms between A and D biased HEB in each organ. As a result, “phenylpropanoid metabolic/biosynthetic process” was enriched in HEB toward the A subgenome in sepals, petals, and stamens, while “response to abiotic stimulus” was over-represented in HEB toward the D subgenome in bracts, sepals, and carpels (Appendix A). The GO term “response to abiotic stimulus” was enriched for HEB toward both subgenomes in petals and stamens (Appendix A).

#### 2.4.2. HEB in MIKC-MADS Box Genes

When HEB was examined across MIKC-MADS box genes, only 41 homoeolog gene pairs were expressed with ≥2 TPM. Among these, TX2094 showed a similar number of HEB toward each subgenome, while TM-1 exhibited more HEB toward the A subgenome (Table 2).

When HEB was investigated in the ABCDE gene classes, a similar number of HEB unique to each accession was observed (32 in TX2094 and 30 in TM-1, Table 3). However, two accessions showed differences in HEB in each class. For example, in TX2094, B and E class genes showed HEB toward the D and A subgenome, respectively, while in TM-1, A and E class genes exhibited HEB toward the A and D subgenomes, respectively (Table 3 and Appendix A). Notably, two accessions did not share HEB in genes related to ovule formation and development (C, D, and B_sister_ genes) in contrast to other class genes (Table 3 and Appendix A). Although the number of homoeologs analyzed is too small to draw definitive biological conclusions, the functional constraints of each floral organ suggest that the two accessions may have evolved differential usage of homoeologs over time. For example, the homologs of A class genes were differentially employed, and their roles in floral meristem formation and flowering regulation [67,68,69] could account for the differential HEB observed in A class gene homoeologs between two accessions. We note the possible relevance of this observation to the later flowering behavior of TX2094 compared to the earlier flowering of TM-1. Also, the difference in seed and fiber development within carpels suggests development programs that differ between the accessions, and which might further explain the non-shared HEB between the two accessions.

### 2.5. Validation of RNA-Seq Results Using Quantitative Real-Time PCR (qRT-PCR)

To verify the accuracy of the transcriptome data, we selected 16 MIKC-MADS box genes, including both homoeologs of seven genes, for qRT-PCR expression analysis. A positive correlation between RNA-Seq and qRT-PCR data is shown in Appendix A. The relative gene expression level of qRT-PCR was largely consistent with the TPM value from the RNA-Seq data, confirming the credibility and accuracy of the RNA-Seq results. However, some discrepancies were observed. For example, the expression levels of the D homoeolog of *AP1.3* were much lower in qRT-PCR compared to RNA-Seq data (Appendix A). Also, the silencing event of *TT16.1* identified by RNA-Seq was not supported by qRT-PCR (Appendix A). These discrepancies may be attributed to the presence of alternative spliced transcripts for *AP1.3* that were not amplified by the primers used in qRT-PCR. Furthermore, the primers used for *TT16.1* in qRT-PCR might lack specificity, potentially amplifying both *TT16* homologs due to their high sequence similarity. This primer specificity issue could also explain inconsistencies in HEB detection between two methods, particularly for *AG* homologs.

## 3. Discussion

The genetic basis of floral development has been widely studied due to its crucial role in plant reproduction. Given the agricultural significance of *G. hirsutum*, research on floral development in cotton has largely focused on individual genes, primarily those involved in carpel and fiber development (e.g., C or D class genes [37,38,39]) or flowering regulation (e.g., A class and *SOC* genes [67,70,71]), as well as genome-wide analyses [42,43]. Many of these studies have specifically investigated MADS-box genes, recognized as key regulators in floral development. To deepen insight into the regulatory mechanisms of floral development in cotton, we analyzed the transcriptomes of each floral organ of both wild (TX2094) and domesticated (TM-1) cotton, with a particular emphasis on MADS-box genes and other key transcription factors.

### 3.1. The ABCDE Model in Gossypium hirsutum

The ABCDE model was derived from the original ABC model, which was proposed to explain floral organogenesis based on mutant phenotypes of *Arabidopsis* and snapdragon [17,18]. Floral organogenesis was explained by the combination of four genes of five classes (the quartet model), and this ABCDE model became generalized and has been supported by many studies across the flowering plants. Previous investigations on cotton have been focused on the expression of these ABCDE genes. For example, de Moura et al. [72] identified five *AG*-like genes (three *AG* and two *STK* homologs) using *G. hirsutum* unigenes and confirmed the expression of three *AG* in stamens and carpels and two *STK* in carpels only, as predicted by the ABCDE model. The availability of the genome sequence of *G. hirsutum* [73] facilitated a genome-wide survey of the MADS-box gene family, and the expression profiles of 11 to 16 selected MADS-box genes were investigated in each floral organ using qRT-PCR [42,43], supporting the ABCDE model in *G. hirsutum*. In addition, recent study of MADS-box protein–protein interaction patterns for AG, STK, SHP, and SEP3 in cotton showed strong conservation in the gene regulatory mechanism for reproductive organ development [74].

In this study, we performed a genome-wide expression analysis for the members of MADS-box gene family using RNA-Seq. In general, MADS-box genes involved in stamen and carpel organogenesis showed agreement with previous studies in *G. hirsutum* as well as other flowering plants. Among ten members of the B class genes, two *AP3*-like genes contain the euAP3 motif, while four *AP3*-like genes have the paleoAP3 motif at the C terminal (Appendix A). It is known that *AP3*s with euAP3 and paleoAP3 motifs confer different functions, with euAP3 involved in petal and stamen differentiation [75]. In our study, *AP3.1*, with the euAP3 motif, exhibited high expression in stamens and carpels, while *AP3.2* also showed similar expression to *AP3.1* in both accessions (Figure 4C,D). However, *AP3.3* exhibited low expression, as supported by the different sequences of the paleoAP3 motif (Appendix A). AP3 and PI function as obligate heterodimers to determine petal and stamen organ identities [76]. Thus, AP3.1 might interact with PI.1 and PI.2 in *G. hirsutum* based on their expression levels as well as the presence of the euAP3 motif (Figure 4C,D). Previous data from floral buds also supported common expression patterns between *AP3.1* (*GhAP3.1.A05*, *GhAP3.1.D05*) and *PI.2* (*GhPI.1.A02*, *GhPI.1.D02*) [43]. In addition, *AP3.2*, with the paleoAP3 motif, may be involved in stamen formation, as its expression was very high in stamens in both RNA-Seq (Figure 4C,D) and qRT-PCR data (Appendix A), as in tomato [77]. However, further experiments should be performed to confirm the functional differentiation between *AP3* genes with euAP3 and paleoAP3 motifs. Two accessions showed differences in expression levels, with TM-1 exhibiting higher expression levels, except for the D homoeolog of *AP3.2*. This variation may be attributed to domestication selection, as stamens, along with carpels (ovules), are associated with domestication traits. Similarly, genes related to carpel development, such as those from the C, D, and B_sister_ classes, displayed comparable patterns (Appendix A).

There are several genes that determine carpel identity, including members of the C, D, and B_sister_ classes. Four, six, and four members of the C, D, and B_sister_ classes, respectively, were identified in cotton, and their expression was restricted to stamens and carpels for C class genes and carpels for D and B_sister_ class genes in both accessions (Figure 5), supporting their conserved function. However, two B_sister_ class genes showed no expression in carpels of TX2094 (Figure 5A) but high expression in TM-1 in agreement with previous studies [42,43]. Since both studies utilized materials from domesticated cotton [42,43], we attempted to confirm the expression of two B_sister_ class genes with qRT-PCR. However, both B_sister_ homologs were expressed in carpels of TX2094 (Appendix A). This could be due to the lack of specificity in the primers used in qRT-PCR or sequence variation in the B_sister_ genes of TX2094, as the primers were designed based on the TM-1 sequences. Further experiments are required to confirm this potential silencing event.

Notably, there was a difference in the expression levels of genes related to carpel and ovule developments between two accessions. In general, those genes showed higher expression in TM-1 (Figure 5B). Although the smaller number of reads was collected from TX2094 (Appendix A), A class genes exhibited much higher expression in TX2094 than TM-1; thus, the expression differences in C, D, and B_sister_ class genes might be explained by the number of ovules, not due to the low reads number in TX2094. It was shown that domesticated cottons have more ovules per fruit [78], and this feature is a significant aspect of the evolutionary and domestication process of the plant [79]. In addition, other parts of the carpels, such as the stigma, style, and ovary, may have diluted the ovule-specific expression in TX2094. This is supported by the observation that *SEP3*, an E class gene forming part of the quartet structure along with AG, SHP, and STK [74], showed relatively similar expression levels between the two accessions (Figure 5C,D). Therefore, further comparative analysis using equivalent carpel parts is necessary to confirm the differential expression between the two accessions.

In contrast to B and C class genes, members of the A/AGL6/E clade showed variable expressions. Based on the ABCDE model, we expected A class gene expression in sepals and petals, but the A class genes were weakly expressed in petals of both accessions (<10 TPM; Figure 3). Eight of twelve A class genes exhibited expression in bracts and sepals of both accessions, as well as in carpels of TX2094. Like B class genes, A class genes have unique motifs, euAP1 and FUL motifs, at the C terminus (Appendix A). Low to no expression of *AP1.1* and *CAL* (Figure 4) might be explained by the absence of these motifs. However, euAP1 and FUL motifs play a minor role in protein function, so their absence may be insignificant [80]. Based on the sequence of the motif (Appendix A) and previous expression work, *AP1.3* might function as an A class gene. However, in contrast to the previous work that showed high expression in sepals and petals (*GhAP1.1* [42]; *GhAP1.11* (Gh_D13G0878) [43]), *AP1.3* was mainly expressed in the bracts, sepals, and carpels of TX2094 and the bracts and sepals of TM-1, but its expression was relatively low in the petals of both accessions (<10 TPM; Figure 4B). Functional analysis of A class genes, such as *GhAP1.1* (*AP1.3*), *GhFUL2* (*FUL.1*) [69], and *AP1.7* (*FUL.1*) [68], revealed their roles in flowering regulation rather than floral organ development. It has been suggested that A class genes are required for sepal identity as well as floral meristem identity, but not petal identity, as a reevaluation of A function has been suggested [55,81]. To assess the reliability of our data, we reanalyzed the petal transcriptome data of a previous study [65]. We confirmed that no A class genes are expressed in petals, while B class genes were highly expressed. Thus, A class genes in *Gossypium* species may not confer petal organ identity but perhaps are involved in floral meristem initiation. If this is the case, *FUL.2* might be involved in floral meristems, as their homologs (*GhFUL1.A07*, *GhFUL1.D07*) were highly expressed in early flowers (the smallest floral buds distinguished from the vegetative buds) among A class genes [43].

E class consists of four *SEP* genes that show redundant functions in floral meristem initiation and floral organogenesis in *Arabidopsis* [82,83]. *Gossypium hirsutum* also contained multiple E class genes, including two *SEP1*, two *SEP2*, four *SEP3*, and four *SEP4* (Appendix A). Except for *SEP4*s, which were mainly expressed in bracts, sepals, and petals, all *SEP* genes were expressed in all floral organs and in developing fibers (Figure 5 and Appendix A). These expression patterns were consistent with the recent study of Chen et al. [84]. They transformed *Arabidopsis* with five *GhSEP* genes, and all transformants resulted in early flowering in a dosage-dependent manner of *GhSEP* protein [44]. Based on protein–protein interaction data of GhSEP with their putative target proteins or DNA motifs, they also proposed the temporal and spatial sequential expression of *SEP* genes towards the inner whorl, from leaf to sepal to petal to stamen to carpel, in an order of *SEP4*, *SEP1*, *SEP2*, and *SEP3* with complex combinations. Our data support their model, as our gene expression patterns match their predicted *SEP* expression, although further investigation should be done on whether all four *GhSEPs* are involved in tetramer formation in petals.

The *AGL6* clade, sister to E class genes, is not a member of the ABCDE model, but it plays an important role in floral development. Functional studies have shown that *AGL6* genes are involved in regulating flowering time, inflorescence, and floral development (reviewed in [85]), but the function of *AGL6* genes remains elusive due to their functional redundancy. In our study, the expression patterns of cotton *AGL6*s, except *AGL6*s without a motif, were similar to those of *SEP2*; both exhibited high expression in sepals and carpels (Figure 5D). This expression pattern was also observed in other core eudicots, including *Actinidia chinensis* [86] and petunia [87], supporting the close relationship between *AGL6* and *SEP* genes based on their sequences (Appendix A). However, their overall expression levels were lower than those of ABCDE class genes in both accessions, implying their role might be less important in floral organogenesis compared to those of ABCDE class genes.

Overall, the ABCDE model partially fits floral organ identity determination in *G. hirsutum.* The genes involved in stamen and carpel formation showed conserved expression patterns following the ABCDE model, while the members of A/E/AGL6 clade exhibited variable expression patterns across bracts and floral tissues along with much lower expression levels compared to those of BCD class genes. Considering the importance of stamens and carpels in reproduction, this result is consistent with the overall level of conservation expected for this key developmental pathway in flowering plants. In contrast, the primary functions of perianth members are often achieved through structural and pigmentation traits rather than relying on precise or tightly coordinated transcriptional control. This flexibility may permit greater variability in gene expression, fostering phenotypic diversity and enabling adaptation to various ecological niches. Notably, the absence of A class gene expression in *Gossypium* petals underscores the need to reevaluate the role of A class genes in cotton and potentially across flowering plants more broadly.

### 3.2. Transcriptome Profiles in Each Floral Organ

Organ-specific transcriptome analysis can enable us to infer the kinds of genes enriched in each organ. Bracts play an important role in plant development and interactions with the environment [88]. For example, bracts are photosynthetically active in cotton and contribute to the yield at the late growth stage [89,90,91]. Functional analysis of the differentially expressed genes also supports bracts having many upregulated genes related to photosynthesis when compared to floral organs, as expected from this green organ (Appendix A). About 62% of biological processes enriched in genes upregulated in bracts compared to floral tissues were related to photosynthesis (Appendix A). STRING also showed strong clusters of genes involved in photosynthesis (Appendix A). Genes upregulated in floral organs, on the other hand, have a diverse range of functions, as expected from the complexity of the developing floral organs.

As the outermost whorl of the flower, sepals protect the inners whorl during floral development, as well as performing photosynthetic activities. Thus, the shared transcriptome profiles between bracts and sepals (Appendix A) may be explained by their shared photosynthetic activities. In both accessions, “response to chemical” emerged as the most significant term in the functional enrichment test. This term encompasses oxygen-containing compounds, such as hormones, signaling molecules, certain metabolites, and reactive oxygen species (ROS) (Appendix A). Given that this term may be over-represented due to stress responses or changes in cellular states, it can be inferred that sepals may play a role in conferring stress response functions in both accessions. Additionally, in TX2094, transcripts upregulated in sepals compared to bracts were also enriched for phenylpropanoid biosynthesis, which serves as a critical molecule for plant responses to abiotic and biotic stresses [92,93]. For example, phenylpropanoid-derived compounds, such as lignin and flavonoids, function in physical stability upon mechanical damage or environmental damage. Thus, transcriptomes of sepals can be differentiated by their protective function compared to those of bracts (Appendix A).

Petals also protect the inner whorls (stamens and carpels) like sepals, but additionally, they attract pollinators with their showy features. In this study, petals share expression profiles with stamens (Appendix A), which is consistent with the ABCDE model (see above). In our functional categorization, petals are differentiated from stamens in being enriched for phenylpropanoid and flavonoid biosynthesis, among other things (Appendix A). In agreement with the functional enrichment test results, a similar number of genes involved in flavonoid biosynthetic pathways were upregulated in petals compared to other organs in TX2094 and TM-1 (29 vs. 28 in Appendix A). Considering that the petals of TX2094 are creamy colored with red petal spots at their base, the enrichment of flavonoid biosynthetic genes is not unexpected. Notably, similar terms were enriched in TM-1, which lacks red petal spots (Appendix A). Considering the diverse functions of flavonoids, including their roles in biotic and abiotic stress responses [94,95], genes involved in flavonoid biosynthesis in TM-1 may be more associated with stress responses rather than pigmentation. Supporting this speculation, genes related to the initial steps of the flavonoid biosynthetic pathways—such as phenylalanine ammonium lyase (*PAL*), cinnamate-4-hydroxylase (*C4H*), and 4-coumaroyl-CoA synthase (*4CL*)—exhibited similar expression patterns between TX2094 and TM-1 (Appendix A). However, genes involved in the later steps of flavonoid biosynthesis, such as dihydroflavonol-4-reductase (*DFR*), anthocyanin synthase (*ANS*), and anthocyanidin reductase (*ANR*), showed higher expression levels in TX2094 compared to TM-1 (Appendix A). Notably, *ANR* homologs, which convert anthocyanins to proanthocyanidin, were highly expressed in the carpels of TX2094 (Appendix A), aligning with the known contribution of proanthocyanidin to brown cotton fibers [96,97,98].

Genes upregulated in stamens of TX2094 were enriched for pathways associated with pollen tube growth and development (Appendix A). Since stamens produce pollen, from which pollen tubes rapidly grow, these pathways are critical for pollen tube growth, particularly as both are involved in cell wall metabolism. In TM-1, GO terms related to cell wall biogenesis, regulation, and modification were enriched (Appendix A). Given that pollen tube growth requires extensive cell wall deposition for rapid expansion [99,100], these enrichment results are consistent. In addition, GO terms related to ethylene metabolic and biosynthetic processes were enriched in both accessions (Appendix A). Ethylene, known to increase during floral organ development and senescence [101,102,103], promotes pollen tube elongation by affecting actin filament organization [104,105,106], supporting its role in facilitating pollen tube growth in both accessions. Furthermore, as ethylene may regulate anther dehiscence in species like tobacco, tomato, and petunia [107,108], it could also play a role in controlling anther dehiscence and pollen release in cotton.

Carpels were characterized by the expression of genes related to cell cycles (Appendix A). Interestingly, PAGE analysis revealed similarities between sepals and carpels, as both exhibited enriched GO terms such as ovule development, gynoecium development, carpel development, and floral whorl development in both accessions (Appendix A). This overlap may be attributed to transcripts that were commonly differentially expressed in sepals and carpels compared to bracts. Approximately 11% of differentially expressed genes were jointly upregulated in both sepals and carpels (Appendix A), including several MADS-box genes such as *AGL6*, *AP1*, *FUL*, *SEP1*, and *SEP2* (Figure 3, Figure 4 and Figure 5). However, carpels in both accessions are distinguished by the enrichment of transcriptions factor families such as *ARF*, *BBR-BPC*, GATA, growth regulating factors (GRF), and ZF-HD (Figure 3B, Appendix A). These transcription factor families are associated with flower development. For instance, GRF is involved in pistil development in *Arabidopsis* [109] and plays a role in the development of gynoecia and anthers [110]. However, some transcription factors still display shared expression patterns between sepals and carpels. For example, *BBR-BPC* homologs were mainly expressed in carpels, but they were also expressed in other floral organs (Figure 7). This is consistent with results from *Arabidopsis*, which has seven *BBR-BPCs* with redundant functions in plant development and growth [111,112]. The quadruple mutants proposed they might be involved in ovule development [111], which was further supported by another study of *BBR-BPC1* [112]. Another interesting feature is that the expression levels of *BBR-PPC* homologs were higher in TM-1 compared to TX2094 (Figure 7). The ovule transcriptome data [66] further support their involvement in ovule development, as their expression levels were highest just before anthesis or on the day of flowering and decreased as the DPA progressed (Figure 7). However, this inference warrants further testing through functional analysis. Similar to *BBR-PPC* homologs, other transcription factors, including GATA, GRF, and MADS box genes (e.g., C and B_sister_ class gene homologs), exhibited comparable expression patterns. Meanwhile, D class gene homologs showed higher expression at later stages of ovule development, such as 10 DPA (Appendix A). D class genes, such as *STK*, are known to specify ovule and seed identity, and their role in modifying the cell wall structure of the seed coat in *Arabidopsis* [113] may explain their distinct expression patterns.

The YABBY family, a plant-specific transcription factor group, plays a critical role in plant growth and development, as its members act as polarity genes [114,115]. Two accessions showed distinct expression patterns: in TX2094, *YABBY* gene homologs were primarily expressed in carpels, with additional expression observed in sepals and bracts, whereas in TM-1, expression was mainly detected in bracts and sepals (Figure 8B). Notably, members of this transcription factor were not enriched in the carpels (ovules) of TM-1 (Appendix A). Since the carpels of two accessions differed in composition—TX2094 carpels included ovules, ovaries, stigma, and style, while TM-1 carpels consisted of ovules only—this suggests that *YABBY* homologs may be expressed in other carpel components, such as the stigma and style, rather than ovules.

### 3.3. HEB of MADS-Box Genes in Wild and Domesticated Cotton

In addition to investigating the ABCDE model in *G. hirsutum*, we examined the HEB of MADS-box genes. This analysis is important because genome merger and doubling can disrupt stoichiometric balance of transcription factors or lead to differential evolutionary trajectories for duplicated genes. Here, we observed HEB toward the D subgenome in the bracts and sepals of TX2094, as well as in all organs analyzed in TM-1 (Figure 2), consistent with previous studies [44,63,64,65]. Interestingly, most HEB patterns were unique to each accession (Table 1), suggesting that their distinct evolutionary histories shaped accession-specific HEB. Notably, TM-1 exhibited a higher number of HEB, perhaps reflecting changes accompanying the domestication process. Supporting this perspective, stamens and carpels shared the least overlap in HEB, suggesting that these organs may have undergone greater differentiation compared to other organs, potentially due to domestication selection, as they are closely associated with domestication traits. Furthermore, 2123 homoeolog gene pairs of transcription factors displayed balanced HEB between subgenomes (Appendix A), indicating that many transcription factors maintained stoichiometric balance despite genome duplication and divergence.

When HEB was analyzed for MADS-box genes, TX2094 displayed a balanced distribution of HEB across both subgenomes, whereas TM-1 exhibited a higher number of HEB toward A subgenome (Table 2). Notably, when accession-specific HEB was considered, TX2094 showed more HEB toward the D subgenome for most genes in the ABCDE model, except for E class genes, which exhibited HEB toward the A subgenome (Appendix A). Interestingly, TM-1 demonstrated the opposite pattern, showing HEB toward the D subgenome for most ABCDE model genes, while E class gene homologs displayed a bias toward the A subgenome (Appendix A). Due to a small number of genes showing HEB toward one subgenome, it is hard to infer what factor affected this difference between TX2094 and TM-1. However, considering shared and unique HEB in each accession, unbalanced HEB was more commonly observed for genes involved in perianth development compared to genes related to stamen and carpel development (Table 3). Thus, functional constraint may explain differences in HEB between TX2094 and TM-1. In addition, although we observed some differential expressions between A and D homoeologs, overall expression patterns were generally similar to each other. Thus, for the most part, conservation of transcription domains appears to be true following genome merger, just as it is between species and families more generally. For example, there was higher expression of the D homoeolog of *AG.2* than that of A homoeolog of *AG.2* in TX2094, or vice versa in TM-1, but both homoeologs showed high expression in stamens and carpels (Figure 5A,B). As we focus on transcription factors, which play a critical role in plant development and reproduction, maintaining both copies in approximate balance seems to be an evolutionary outcome consistent with strong selection for conservation of function. Also, as we only consider homoeolog-specific reads for their expression to compare them, their differential expression will not reflect their true expression difference between A and D homoeologs. Thus, we only attest that there is no silencing of either homoeolog among MADS-box genes in floral organs, but it is hard to reach to a conclusion about their relative contribution to floral development. Further investigations into epigenetic features, such as histone modifications, and post-transcriptional regulation mechanisms via small RNAs will be valuable for understanding the differential HEB between wild and domesticated cottons, as these factors seem to play a key role in HEB in cotton [116,117,118,119].

## 4. Materials and Methods

### 4.1. Plant Materials and Library Construction

A wild form of *G. hirsutum* accession TX2094, originally from Yucatan, Mexico [120], was sampled in two replicates from two individual trees. Each replicate consisted of three flowers. Each flower was collected on the day of flowering around 11:00 AM and immediately dissected to harvest bracts, sepals, petals, stamens (including anthers and filaments), and carpels (including stigma, style, ovary, and ovules) at 0 DPA, which were snap-frozen in liquid nitrogen until extraction. Floral buds *of G. arboreum* and *G. raimondii*, representing models of the diploid progenitors (A and D, respectively) of allopolyploid cotton, were also collected as controls. Total RNAs were extracted using a CTAB extraction protocol [121] and purified using the RNeasy Plant Mini Kit (Qiagen, Stanford, CA, USA). Purified RNAs were quantified and qualified with Agilent 2100 Bioanalyzer (Agilent, Santa Clara, CA, USA), and mRNAs were purified using the MicroPoly(A) Purist kit (Ambion, Austin, TX, USA). A total of twelve RNA-Seq libraries (two biological replicates X five tissue types, two with floral buds) were constructed using the NEBNext mRNA Sample Prep Master Mix Set 1 following the manufacturer’s suggestion (New England Biolabs, Ipswich, MA, USA). The constructed libraries were indexed with six nucleotide sequences and pooled in equimolar amounts. Pooled libraries were sequenced on the Illumina HiSeq 2000 sequencer with single-end 100 base reads at the Genomics Core Facility at the University of Oregon. Short read sequences were deposited in the NCBI Sequence Read Archive (SRA) under the accession number PRJNA929459. In addition, we reanalyzed RNA-Seq data from seven organs—bracts, sepals, petals, anthers, filaments, ovules at −3, 0, 1, 3, and 10 DPA, fibers at 10 DPA—of *G. hirsutum* cv. TM-1 (NCBI SRA accession no. PRJNA490626) [66]. Data from anthers and filaments were combined as stamens, while data from ovules at 0 DPA represented carpels in this study.

### 4.2. Analysis of RNA-Seq Data

Raw reads were sorted according to their indexed nucleotides. After trimming the indexed sequences, reads were filtered based on the quality scores (Q ≥ 20) and read length (≥17 bp) with a fastx tool kit (https://github.com/agordon/fastx_toolkit, accessed on 1 April 2024.). Fastq formatted reads were mapped to the reference genome (*Gossypium hirsutum* v2.1; 75,376 protein-coding loci) [44] using BWA [122]. The DESeq2 package (ver. 1.28.1) was used to detect differentially expressed transcripts in each floral organ relative to bract [123], and differential expression was defined when a transcript showed at least 2-fold change with TPM ≥ 5 in all biological replicates of at least one organ type. The distribution of *p*-values was controlled for a false discovery rate (FDR) by the BH method [124] at α = 0.05. Differentially expressed transcripts, log2 transformed (log2 (TPM + 1)), were clustered using *k*-means clustering with an R package “pheatmap”.

Homoeologous gene pairs were identified using BLAST v.2.12.0 [125,126] and CottonGen (accessed on 1 April 2024) [127] and confirmed with their length, chromosomal positions, and functional annotation. As a result, 22,944 homoeologous gene pairs were used in evaluating HEB. The same methods and criteria described above were applied to identify differential HEB, but uniquely mapped reads were selected for this analysis. That is, the raw read number was analyzed with DESeq2, and HEB was determined with at least 2-fold change with TPM ≥ 5 in all biological replicates of at least one organ type (adjusted *p*-value or FDR < 0.05).

To explore the nature of the biological pathways involved in floral development, differentially expressed genes in each floral organ relative to bract were analyzed using the Parametric Analysis of Gene set Enrichment (PAGE) tool of agriGO (http://systemsbiology.cau.edu.cn/agriGOv2/, accessed on 30 May 2024) [128,129]. For PAGE, we used transcripts identified as differentially expressed (DE) between bracts and each floral organ, with multi-test adjustment of the Benjamini–Yekutieli method [130], and a minimum ten mapping entries. We also analyzed DE transcripts in terms of protein network using STRING which implements Gene Ontology (GO), KEGG pathways, and new classification systems based on high-throughput text mining [131]. To investigate the interaction of transcription factors differentially expressed in each floral organ relative to bracts, *k*-means clustering was performed. The number of clusters was determined by the software, and the cluster with the largest number of genes was further analyzed using MCL clustering on STRING.

Fisher’s Exact test [132] was performed to determine whether specific transcription factor families were enriched in floral organs compared to the subtending floral bracts. For transcription factors identified as enriched in floral organ(s), we investigated the expression patterns of individual paralogs and homoeologs based on their phylogenetic relationships. We first retrieved sequences based on annotation but also explored the entire genome by homology search using *Arabidopsis thaliana* homologs with BLAST [125,126]. Retrieved sequences were aligned via Clustal W [133], and phylogenetic trees were constructed using MEGA X (version 10.1.7) with a default option of Maximum likelihood [134].

### 4.3. Validation of RNA-seq Results Using qRT-PCR

To validate RNA-Seq results, we employed qRT-PCR on 16 genes belonging to the ABCDE class (Appendix A). RNAs were extracted as described above, and their quality and quantity were assessed using a NanoDrop™ One Microvolume UV-Vis Spectrophotometer (Thermo Fischer Scientific, Waltham, MA, USA) and agarose gel electrophoresis. cDNA was synthesized using the Maxima First Strand cDNA Synthesis Kit for RT-qPCR with dsDNase (Thermo Fischer Scientific, Waltham, MA, USA). qRT-PCR was conducted using the Luna^®^ Universal qPCR Master Mix (New England Labs, Ipswich, MA, USA) on a CFX96 Touch Real-Time PCR Detection System (Bio-Rad, Hercules, CA, USA). Each reaction was performed with three biological replicates, each consisting of two flowers. The relative expression of target genes was calculated using the comparative C_t_ method (Applied Biosystems, Framingham, MA, USA), with *GhACT7* as the internal control.

## 5. Conclusions

This study investigated gene expressions in the flowers of *G. hirsutum* using genome-wide transcriptome profiling across each floral organ. We characterized the expression of MIKC-type MADS-box genes and other transcription factors. As expected, the ABCDE model largely aligns with floral organogenesis in cotton, except for A class genes, suggesting the need for a reevaluation of A function genes in floral organogenesis. Organ-specific transcriptome analysis revealed distinct functional enrichment in each organ. For example, sepals showed enrichment in photosynthetic and protective functions, while petals were enriched for phenylpropanoid and flavonoid synthesis pathways involved in pigmentation and stress responses. In stamens, upregulated genes were enriched for pathways associated with pollen tube growth, actin and cytoskeleton organization, and cell wall metabolism. Carpels were characterized by genes related to cell cycles, carpel (ovule) identity, and seed formation. While several transcription factors were enriched in specific floral organs, many were expressed in both vegetative and reproductive tissues, suggesting their broader involvement in plant growth and development. Homoeolog specific gene expression analysis revealed a D subgenome bias in certain organs, including bracts, sepals, and carpels (ovules). The ABCDE class genes exhibited differential subgenome biased HEB between TX2094 and TM-1, suggesting that domestication processes have influenced homoeolog utilization despite functional constraints in floral organogenesis. Further analysis is needed to explore how subgenome-biased HEB impacts the floral development of cotton.

## Figures and Tables

**Figure 1 plants-14-00502-f001:**
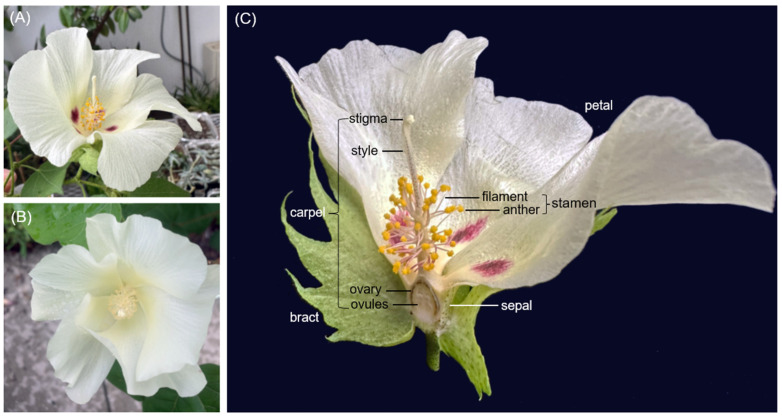
The flower of *G. hirsutum.* (**A**) TX2094, (**B**) domesticated *G. hirsutum* Maxxa, which has a flower similar to TM-1, (**C**) floral organs of TX2094. For the carpel, all parts, including stigma, style, ovary, and ovules, were used for TX2094, while only ovules at 0 DPA were employed for TM-1. (Photo: M. Yoo).

**Figure 2 plants-14-00502-f002:**
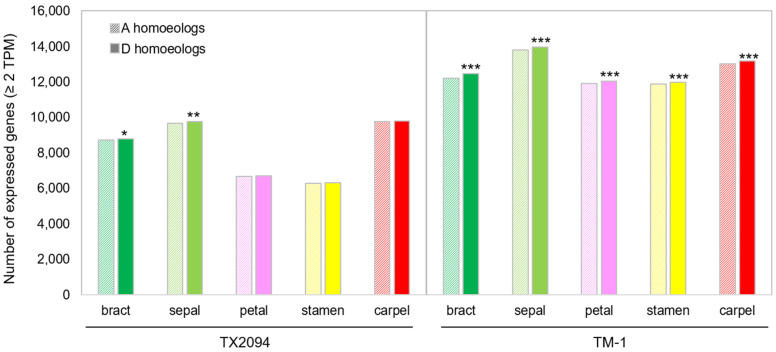
Expression levels of homoeologs were compared among different organs in each accession. The number of homoeologous genes that are expressed (≥2 TPM) in the A or D subgenome are shown. * *p*-value < 0.5, ** *p*-value < 0.1, *** *p*-value < 0.01 from two-tailed binomial test.

**Figure 3 plants-14-00502-f003:**
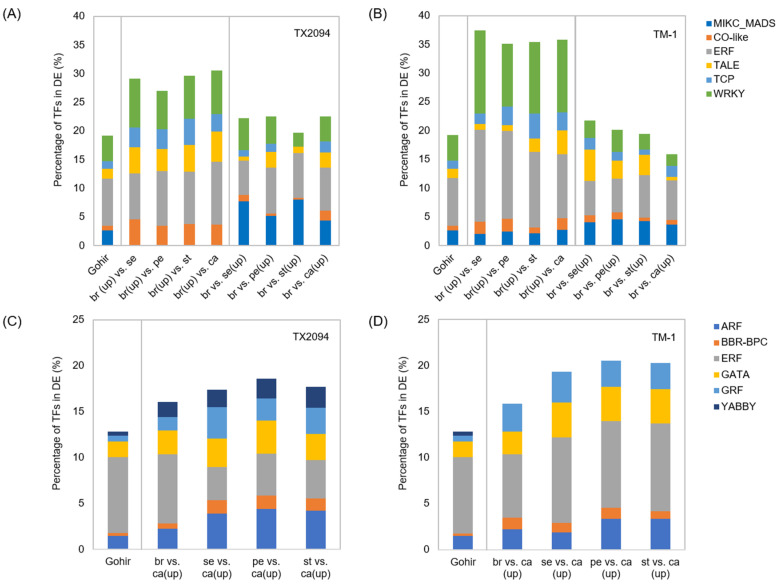
Enriched transcription factors (TFs) in downregulated and upregulated transcripts in bracts compared to floral organs, respectively, in TX2094 (**A**) and TM-1 (**B**), and upregulated transcripts in carpels compared to other organs in TX2094 (**C**) and TM-1 (**D**). Gohir (%) shows the percentage of TFs from the reference database. br: bracts, se: sepals, pe: petals, st: stamens, ca: carpels.

**Figure 4 plants-14-00502-f004:**
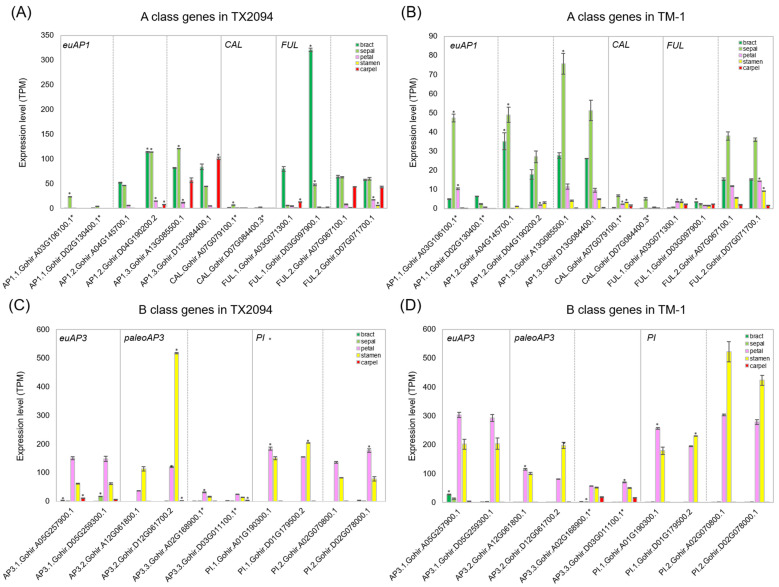
Expression patterns of selected MIKC genes. (**A**) A class gene homologs in TX2094, (**B**) A class gene homologs in TM-1, (**C**) B class gene homologs in TX2094, (**D**) B class gene homologs in TM-1. Asterisks (*) next to gene IDs and on the bars denote transcripts lacking a clade-specific motif and exhibiting higher expression relative to their homoeologs (adjusted *p*-value < 0.05), respectively.

**Figure 5 plants-14-00502-f005:**
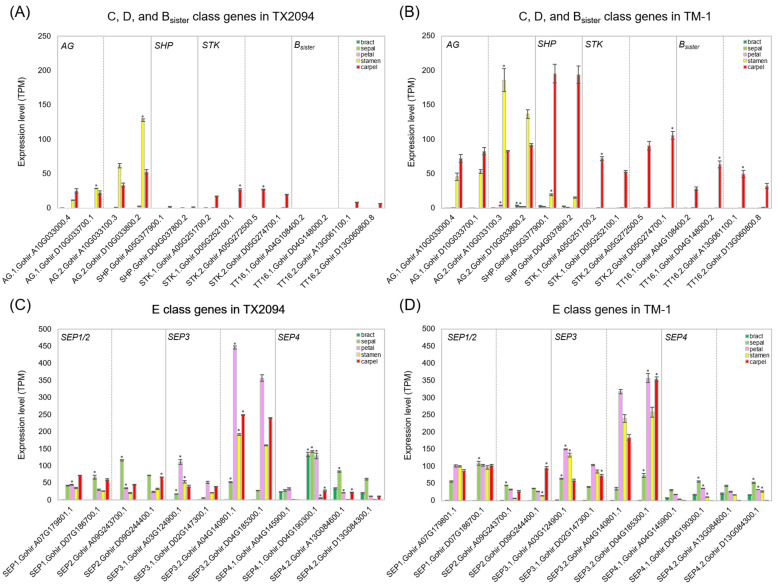
Expression patterns of selected MIKC genes. (**A**) C, D, and B_sister_ class gene homologs in TX2094, (**B**) C, D, and B_sister_ class gene homologs in TM-1, (**C**) E class gene homologs in TX2094, (**D**) E class gene homologs in TM-1. Asterisk (*) on the bars denotes exhibiting higher expression relative to their homoeologs (adjusted *p*-value < 0.05).

**Figure 6 plants-14-00502-f006:**
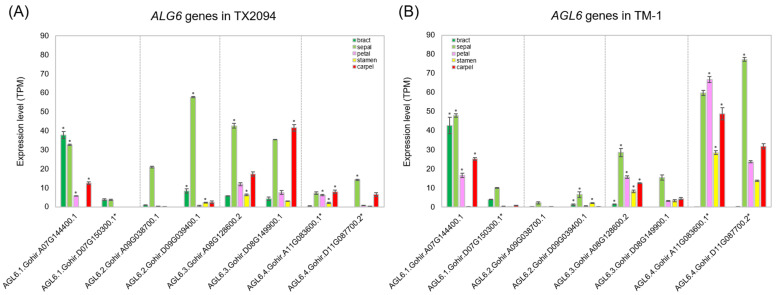
Expression patterns of selected MIKC genes. (**A**) *AGL6* gene homologs in TX2094, (**B**) *AGL6* gene homologs in TM-1. Asterisks (*) next to gene IDs and on the bars denote transcripts lacking a clade-specific motif and exhibiting higher expression relative to their homoeologs (adjusted *p*-value < 0.05), respectively.

**Figure 7 plants-14-00502-f007:**
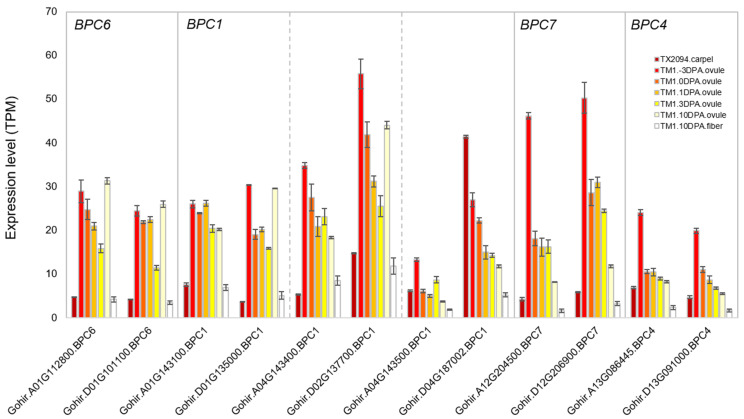
Expression patterns of *BBR-BPC* gene homologs in carpels and ovules of TX2094 and TM-1.

**Figure 8 plants-14-00502-f008:**
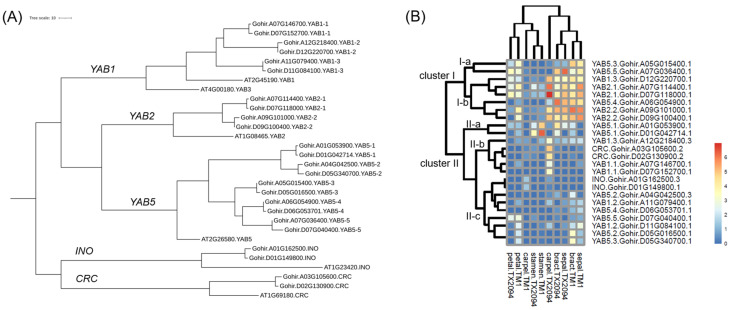
Expression patterns of YABBY gene homologs. (**A**) Maximum likelihood tree of *G. hirsutum* YABBY gene homologs. (**B**) Hierarchical clustering of expressions of *G. hirsutum* YABBY gene homologs. Log2-transformed TPM values (log2(TPM + 1)) were used to generate the heatmap. The color scale bar in the bottom-right corner indicates high-expression (red) and no expression (blue).

**Figure 9 plants-14-00502-f009:**
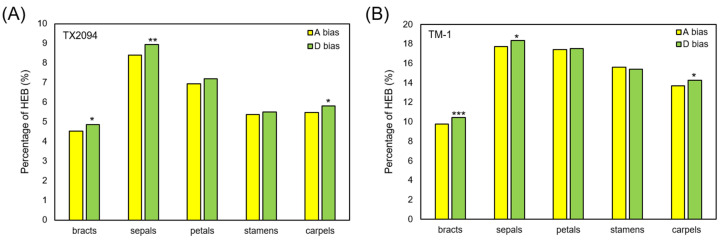
The percentage of homoeolog expression bias in each organ of TX2094 (**A**) and TM-1 (**B**). * *p*-value < 0.05, ** *p*-value < 0.1, *** *p*-value < 0.001 from a proportion test between two subgenome biases.

**Table 1 plants-14-00502-t001:** Homoeolog expression bias identified in each organ. The percentage was calculated against the total number of HEB in each organ of both TX2094 and TM-1.

Category	Bias	Bracts	Sepals	Petals	Stamens	Carpels
TX2094_only	A	510	622	395	440	604
	D	508	586	390	503	623
TM1_only	A	1704	2711	2737	2727	2448
	D	1742	2691	2702	2726	2523
Shared	A	440 (7.9%)	1151 (12.4%)	1072 (12.5%)	716 (9.1%)	542 (7.2%)
	D	537 (9.7%)	1311 (14.1%)	1132 (13.2%)	675 (8.6%)	595 (7.9%)
Changes from TX2094 to TM-1	A→D	62	106	89	49	78
	D→A	44	104	86	55	82

**Table 2 plants-14-00502-t002:** Homoeolog expression bias identified in each organ. The percentage was calculated against the total number of HEB in each organ of both TX2094 and TM-1. * Total = 41 homoeologous gene pairs × 5 organs.

	A > D	A < D	A = D	Low Expression (<2 TPM)	Total *
TX2094	50	46	29	80	205
TM-1	49	34	34	88	205

**Table 3 plants-14-00502-t003:** Homoeolog expression bias identified in the ABCDE class genes. A and D bias indicate HEB toward A and D subgenome, respectively.

MIKC-MADS Box Genes	HEB in TX2094 Only		HEB in TM-1 Only		Shared HEB	
	A Bias	D Bias	A Bias	D Bias	A Bias	D Bias
A class	3	5	7	0	2	4
B class	2	5	2	1	2	2
C/D/B_sister_ class	1	3	5	4	0	0
E class	10	2	0	8	4	5
*AGL6*	0	1	3	0	9	4
Total	16	16	17	13	18	14

## Data Availability

Short read sequences were deposited in the NCBI Sequence Read Archive (SRA) under the accession number PRJNA929459.

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
