# Peer review of "Comparative Analysis of Floral Transcriptomes in Gossypium hirsutum (Malvaceae)"

_plants, 2025, doi:10.3390/plants14040502_

Round 1

Reviewer 1 Report

Comments and Suggestions for Authors

Title: Comparative analysis of floral transcriptomes in Gossypium 2 hirsutum (Malvaceae)

        Summary: Nobles et al. analyzed the transcriptomes of floral organs and bracts in wild and domesticated Gossypium hirsutum, specifically emphasizing the MADS-box genes and their roles in floral development. Several interesting results were found, including domesticated cotton exhibiting higher transcript accumulation in carpel than wild type. Photosynthesis-related genes were enriched in bracts, while the floral organs exhibited showed the expression of a wide variety of genes.

Although the article offers valuable insights for advancing gene expression activities aimed at enhancing fiber and cottonseed development, its current form is limited and needs modifications, along with the supplementation of additional results.

My recommendations are listed below.

1.        Title: Why did authors prioritize MADS-box Genes over other flower development genes such as LEAFY (LFY), APETALA2-like Genes, SUPPRESSOR OF OVEREXPRESSION OF CONSTANS1 (SOC1), FLOWERING LOCUS T (FT), CONSTANS (CO), AGL6-like Genes, WUSCHEL (WUS), etc.

2.        Line 25:meaningfully” Please replace with a more suitable alternative

3.        Line 40: Please insert a citation before the full stop

4.        Figure 1: How did authors compute significance values without adding the standard deviations on the bar?

5.        Line 104: “…….. which an average of 72.48% were mapped onto the reference genome of G. hirsutum.”

This sentence sounds a bit ambiguous. Is 72.48% really an average of 76,018,537 ?

6.        Line 106:Among transcripts with Transcripts Per Kilobase …………………… (Table S2).”

Please state the numbers and put the percentages in parentheses

7.        Line 111: Could authors be more specific “sample differences” look through broad. Authors should consider stating the specific difference

8.        Figure S1 is too complicated to follow and difficult to follow. Apart from it not being fully described in the text, the legend failed to describe the numbers in the parenthesis.

9.        Line 135, 149 etc.: “each contrast”? Authors should be specific. There is widespread of this issue in the manuscript.

10.      Figure S2: Please consider placing the sample names below the heatmap. The dendrogram masks some labelling.

11.      A transcriptional regulatory network and TF identified could clarified the interaction among the genes identify. Author may consider using STRING and Cytoscape to perform this analysis

12.      Showing the TPM levels alone is not enough, authors need to validate their results using qPCR.

13.      Authors should consider showing photos of the floral parts used for the transcriptome analysis

14.      I strongly recommend authors to show by model, how various key pathways, genes, or TFs identified possibly interact to execute specific functions.

15.      Cotton fiber and seed development benefits from floral gene expression as the two are crucially linked. I suggest authors add at least 2 sentences on the importance of cotton fibers and seed in their introduction.

Comments on the Quality of English Language

Generally looks good but few errors requires attention

Author Response

Dear Reviewers,

We sincerely thank you for taking the time to provide valuable feedback on our manuscript. Your thoughtful comments and suggestions have significantly improved the quality and clarity of our work.

We have carefully addressed each of your points and revised the manuscript accordingly. A detailed response to your comments has been provided, with changes highlighted in red text and additional explanations included where necessary.

We greatly appreciate your efforts in helping us refine our work and hope that the revisions meet your expectations. Please do not hesitate to reach out if further clarifications are needed.

Thank you again for your insights and contributions.

Sincerely,
Mi-Jeong Yoo

Response to Reviewer 1’s comments

Title: Comparative analysis of floral transcriptomes in Gossypium 2 hirsutum (Malvaceae)

 Summary: Nobles et al. analyzed the transcriptomes of floral organs and bracts in wild and domesticated Gossypium hirsutum, specifically emphasizing the MADS-box genes and their roles in floral development. Several interesting results were found, including domesticated cotton exhibiting higher transcript accumulation in carpel than wild type. Photosynthesis-related genes were enriched in bracts, while the floral organs exhibited showed the expression of a wide variety of genes.

Although the article offers valuable insights for advancing gene expression activities aimed at enhancing fiber and cottonseed development, its current form is limited and needs modifications, along with the supplementation of additional results.

My recommendations are listed below.

  1. Title: Why did authors prioritize MADS-box Genes over other flower development genes such as LEAFY (LFY), APETALA2-like Genes, SUPPRESSOR OF OVEREXPRESSION OF CONSTANS1 (SOC1), FLOWERING LOCUS T (FT), CONSTANS (CO), AGL6-like Genes, WUSCHEL (WUS), etc.

=> Thank you for your comment. We analyzed the transcriptomes of individual floral organs with high MADS-box gene expression. The genes highlighted by the reviewer primarily play a role in floral meristem development during the early stages of organ formation. However, their expression levels in the organs we examined were relatively low (<20 TPM) compared to those of MADS-box genes, particularly the ABCDE genes. Nonetheless, we referenced SOC1 in lines 321–323 and included AGL6 homologs, as they are closely related to E class genes.

  1. Line 25: “meaningfully” Please replace with a more suitable alternative

=> We replaced “meaningfully” with “significantly” [Line 25].

  1. Line 40: Please insert a citation before the full stop.

=> The following two papers were cited [Line 41].

  • De Bodt, S., Raes, J., Florquin, K., Rombauts, S., Rouze, P., Theissen, G., and Van De Peer, Y. (2003). Genomewide structural annotation and evolutionary analysis of the type I MADS-box genes in plants. J Mol Evol 56, 573-586. 10.1007/s00239-002-2426-x.
  • Gramzow, L., and Theissen, G. (2010). A hitchhiker's guide to the MADS world of plants. Genome Biol 11, 10.1186/gb-2010-11-6-214.

  1. Figure 1: How did authors compute significance values without adding the standard deviations on the bar?

=> We tested whether more D homoeologs were expressed than A homoeologs using a two-tailed binomial test; therefore, standard deviation bars are not required. We revised the figure caption [Lines 143 – 144]

  1. Line 104: “…….. which an average of 72.48% were mapped onto the reference genome of G. hirsutum.” This sentence sounds a bit ambiguous. Is 72.48% really an average of 76,018,537?

=> Now this sentence was revised with more clarity as below as 72.48% of 76,018,537 is 55,101,045 (5.5 M reads per replicate).

Lines 117 – 119: A total of 76,018,537 clean reads were obtained from bracts and floral organs, and on average, 72.48% of these reads were successfully mapped onto the reference genome of G. hirsutum.

  1. Line 106: “Among transcripts with Transcripts Per Kilobase …………………… (Table S2).”

Please state the numbers and put the percentages in parentheses

=> We included the numbers with the percentages in parenthesis (Line 121).

  1. Line 111: Could authors be more specific “sample differences” look through broad. Authors should consider stating the specific difference

=> We added Figure 1 to highlight the difference of floral sample between TX2094 and TM-1 (Lines 131 – 135)

Figure 1. The flower of G. hirsutum. (A) TX2094, (B) domesticated G. hirsutum, (C) floral organs of TX2094. For the carpel, all parts, including stigma, style, ovary, and ovules, were used for TX2094, while only ovules at 0 DPA were employed for TM-1. (Photo: M. Yoo)

  1. Figure S1 is too complicated to follow and difficult to follow. Apart from it not being fully described in the text, the legend failed to describe the numbers in the parenthesis.

=> We modified Figure S1 and added more explanation in the legend of Figure S1 as below.

Figure S1: Number of differentially expressed (DE) genes between organs of (A) TX2094 and (B) TM-1. The highest and lowest number of DE genes are shown in red and blue text, respectively. The organ-specific breakdown is shown below the total DE number. For example, in the comparison between carpel and bract, a total of 12688 DE genes were identified. Of these, 6605 and 6085 genes were more abundant in carpel and bract, respectively. br: bract, se: sepal, pe: petal, st: stamen, ca: carpel.

  1. Line 135, 149 etc.: “each contrast”? Authors should be specific. There is widespread of this issue in the manuscript.

=> We revised “each contrast” to “each floral organ relative to bract” across the manuscript.

  1. Figure S2: Please consider placing the sample names below the heatmap. The dendrogram masks some labelling.

=> We placed the sample names below the heatmap in Figure S2.

  1. A transcriptional regulatory network and TF identified could clarify the interaction among the genes identify. Author may consider using STRING and Cytoscape to perform this analysis.

=> We provided STRING results to investigate transcriptional regulatory network as Figure S6 and revised the text in lines 229 – 234.

  1. Showing the TPM levels alone is not enough, authors need to validate their results using qPCR.

=> We added qRT-PCR data to supplement RNA-Seq data in Results (section 2.5) and Methods (section 4.3).

  1. Authors should consider showing photos of the floral parts used for the transcriptome analysis

=> We added Figure 1 showing the floral parts used in this study.

  1. I strongly recommend authors to show by model, how various key pathways, genes, or TFs identified possibly interact to execute specific functions.

=> We appreciate your comment. However, as shown in STRING analysis (Figures S6), various pathways are involved in each floral organ formation with the strongest interaction among MADS-box genes. Thus, it is hard to provide a specific model for each floral organ formation based on our expression data.

  1. Cotton fiber and seed development benefits from floral gene expression as the two are crucially linked. I suggest authors add at least 2 sentences on the importance of cotton fibers and seed in their introduction.

=> We revised our introduction based on reviewer’s suggestion [Lines 80 – 88].

Thank you again for your valuable comments!

Reviewer 2 Report

Comments and Suggestions for Authors

The author conducted research on the "Comparative analysis of floral transcriptomes in Gossypium hirsutum (Malvaceae)." Overall, the paper has certain research value and is written fluently, but there are some issues.

1.     Lines 36-37 "MADS-box genes are key players in floral development regulation as well as other developmental processes in plants [1-11]." This is an excessive citation. A single sentence encompasses 11 references, which either need to be specified or reduced.

2.     Lines 52-56 The content here is repetitive. The author needs to carefully review and make corrections.

3.     Lines 80-81 "Thus, studies have been performed on some MADS-box genes in Gossypium species [43-47]." What are the specific details? There are five references in this short text.

4.     The author should provide background information on the wild form of G. hirsutum accession TX2094 and TM-1 used in the text in the introduction.

5.     How was the significant difference in Figure 1 determined? Based on what statistical method and what samples were used for the statistical analysis? The figure caption should indicate this.

6.     The author should provide images of the Gossypium hirsutum to give readers an intuitive understanding.

7.     Figure 3, how were the p-values obtained? If based on TPM, how can TPM be used for comparison? TPM is a normalization metric that reflects the proportion of transcripts in the total expression of the sample, not absolute read counts. Since the total expression of the sample (the sum of TPMs of all transcripts) is normalized to a million,: if the expression of some genes increases significantly, it can suppress the TPM values of other genes, even if their original read counts have not changed. This dependency violates the independence assumption required for statistical tests. The proportional nature and lack of independence make it unsuitable for statistical analysis.

8.     How was the heatmap visualization in Figure 7B normalized?

9.     414-415 "Given the agricultural significance of G. hirsutum, research on floral development in cotton has focused extensively on individual genes [44,45,72-77]." This needs to be more specific; excessive citations are present. A single sentence contains 7 references, making it difficult for readers to extract meaningful information.

10.   537-549 The function of Class A genes in Gossypium (cotton) petals—are there any experimental results supporting these findings? If so, they should be included in the discussion. Alternatively, it is recommended to explore, based on existing refs, whether there might be other regulatory factors compensating for the function of Class A genes.

11.   The current data primarily rely on RNA-Seq results. It is recommended to supplement these with qPCR results.

Author Response

Dear Reviewers,

We sincerely thank you for taking the time to provide valuable feedback on our manuscript. Your thoughtful comments and suggestions have significantly improved the quality and clarity of our work.

We have carefully addressed each of your points and revised the manuscript accordingly. A detailed response to your comments has been provided, with changes highlighted in red text and additional explanations included where necessary.

We greatly appreciate your efforts in helping us refine our work and hope that the revisions meet your expectations. Please do not hesitate to reach out if further clarifications are needed.

Thank you again for your insights and contributions.

Sincerely,
Mi-Jeong Yoo

Response to Reviewer 2’s comments

The author conducted research on the "Comparative analysis of floral transcriptomes in Gossypium hirsutum (Malvaceae)." Overall, the paper has certain research value and is written fluently, but there are some issues.

  1. Lines 36-37 "MADS-box genes are key players in floral development regulation as well as other developmental processes in plants [1-11]." This is an excessive citation. A single sentence encompasses 11 references, which either need to be specified or reduced.

=> We cited only the three most recent and significant papers [Line 38].

  1. Lines 52-56 The content here is repetitive. The author needs to carefully review and make corrections.

=> We revised this part as follows.

Lines 53 – 56: Based on this model, the combination of the ABCDE class genes determine each floral organ identity as follows: sepals (A + E), petals (A + B + E), stamens (B + C + E), carpels (C + E), and ovules (D + Bsister + E) through the formation of unique protein tetramers [17,22-25].

  1. Lines 80-81 "Thus, studies have been performed on some MADS-box genes in Gossypium species [43-47]." What are the specific details? There are five references in this short text.

=> We clarified each reference and revised this section as follows.

Lines 81 – 89: Consequently, research has primarily focused on MADS-box genes involved in carpel and fiber development, such as the AG homolog (GhMADS3) [37], SHATTERPROOF (SHP; GhMADS7) and SEEDSTICK (STK; GhMADS5, GhMADS6) homologs, which exhibit high expression in carpels and developing fibers, [38,39]. Additionally, the AP3 homolog (GhMADS9) [40], which is related to petal and stamen organ identify, has been studied. Jiang et al. [41] identified and analyzed the expression of 53 MIKC-type genes across various organs, including flowers, individual floral organs, fibers, and ovules, revealing expression patterns broadly consistent with those observed in Arabidopsis.

  1. The author should provide background information on the wild form of G. hirsutum accession TX2094 and TM-1 used in the text in the introduction.

=> We provided background information on the wild form of G. hirsutum accession TX2094 and TM-1 [Lines 102 - 107].

  1. How was the significant difference in Figure 1 determined? Based on what statistical method and what samples were used for the statistical analysis? The figure caption should indicate this.

=> We tested whether more D homoeologs were expressed than A homoeologs using a two-tailed binomial test, and this was added to the Figure caption [lines 143 – 144].

  1. The author should provide images of the Gossypium hirsutum to give readers an intuitive understanding.

=> We added the flowers of both accessions in Figure 1.

  1. Figure 3, how were the p-values obtained? If based on TPM, how can TPM be used for comparison? TPM is a normalization metric that reflects the proportion of transcripts in the total expression of the sample, not absolute read counts. Since the total expression of the sample (the sum of TPMs of all transcripts) is normalized to a million: if the expression of some genes increases significantly, it can suppress the TPM values of other genes, even if their original read counts have not changed. This dependency violates the independence assumption required for statistical tests. The proportional nature and lack of independence make it unsuitable for statistical analysis.

=> We apologize for the confusion; this was an oversight on our part. We determined HEB between homoeolog pairs based on an adjusted p-value < 0.05, as calculated by DESeq2, which was used for differential expression analysis in homoeolog-specific data. Figure captions were properly revised from p-value to adjusted p-value.

  1. How was the heatmap visualization in Figure 7B normalized?

=> The heatmap was generated using log2(TPM + 1), indicating that the data had been normalized using TPM [Line 374, 784]

  1. 414-415 "Given the agricultural significance of G. hirsutum, research on floral development in cotton has focused extensively on individual genes [44,45,72-77]." This needs to be more specific; excessive citations are present. A single sentence contains 7 references, making it difficult for readers to extract meaningful information.

=> We revised this part as follows [lines 463 – 467].

Given the agricultural significance of G. hirsutum, research on floral development in cotton has largely focused on individual genes, primarily those involved in carpel and fiber development (e.g., C or D class genes [37-39]) or flowering regulation (e.g., A class and SOC genes [66-68]), as well as genome-wide analyses [42,43,69].

  1. 537-549 The function of Class A genes in Gossypium (cotton) petals—are there any experimental results supporting these findings? If so, they should be included in the discussion. Alternatively, it is recommended to explore, based on existing refs, whether there might be other regulatory factors compensating for the function of Class A genes.

=> We added functional studies of A class gene in lines 551 – 553.

  1. The current data primarily relies on RNA-Seq results. It is recommended to supplement these with qPCR results.

=> We added qRT-PCR data to supplement RNA-Seq data in Results (section 2.5) and Methods (section 4.3).

Thank you again for your valuable comments!

Reviewer 3 Report

Comments and Suggestions for Authors

Summary

This manuscript presents a comparative analysis of floral transcriptomes in wild and domesticated Gossypium hirsutum (cotton). The study focuses on MADS-box genes and other transcription factors involved in floral development, using RNA-Seq data from different floral organs and bracts. The authors investigate the expression patterns of ABCDE class genes, analyze differential gene expression between organs, and explore functional enrichment of differentially expressed transcripts.

Major comments

1. More details about the samples, including biological replicates and developmental stages, are needed to assess the robustness of the results.

2. The manuscript lacks information on statistical methods used for differential expression analysis and functional enrichment tests. This information is essential for evaluating the validity of the results.

3. While mentioned in the introduction, the results do not clearly present how the ABCDE model was evaluated using genome-wide data. This analysis should be more explicitly described and discussed.

4. The comparison of A and D homoeolog expression levels across floral organs, mentioned as an aim, is not clearly presented in the results. This analysis should be expanded and highlighted.

Minor comments

1. Some figures, particularly Figure 1, could be improved for clarity. Consider using color-coding or different symbols to distinguish between accessions and organs.

2. There is some repetition in the results section, particularly regarding differential expression between organs. This could be condensed for better readability.

3. Some technical terms (e.g., HEB, DPA) are used without explanation. These should be defined upon first use.

4. Consider using a table to summarize the key differentially expressed transcription factors in each organ, rather than listing them in the text.

The manuscript is scientifically sound and makes a significant contribution to understanding floral transcriptomics in cotton. With minor revisions, it is suitable for publication in Plants.

Author Response

Dear Reviewers,

We sincerely thank you for taking the time to provide valuable feedback on our manuscript. Your thoughtful comments and suggestions have significantly improved the quality and clarity of our work.

We have carefully addressed each of your points and revised the manuscript accordingly. A detailed response to your comments has been provided, with changes highlighted in red text and additional explanations included where necessary.

We greatly appreciate your efforts in helping us refine our work and hope that the revisions meet your expectations. Please do not hesitate to reach out if further clarifications are needed.

Thank you again for your insights and contributions.

Sincerely,
Mi-Jeong Yoo

Response to Reviewer 3’s comments

This manuscript presents a comparative analysis of floral transcriptomes in wild and domesticated Gossypium hirsutum (cotton). The study focuses on MADS-box genes and other transcription factors involved in floral development, using RNA-Seq data from different floral organs and bracts. The authors investigate the expression patterns of ABCDE class genes, analyze differential gene expression between organs, and explore functional enrichment of differentially expressed transcripts.

Major comments

  1. More details about the samples, including biological replicates and developmental stages, are needed to assess the robustness of the results.

=> We presented the photos of two accessions as Figure 1 and added more details on the material [lines 126 – 128].

  1. The manuscript lacks information on statistical methods used for differential expression analysis and functional enrichment tests. This information is essential for evaluating the validity of the results.

=> We included statistical test information in the figures which present the expression data and provided more details in the Methods section.

  1. While mentioned in the introduction, the results do not clearly present how the ABCDE model was evaluated using genome-wide data. This analysis should be more explicitly described and discussed.

=> Thank you for your comment. We utilized genome-wide data to include all members of the ABCDE model and analyzed their expression using transcriptomic profiles of each floral organ. The overall expression patterns presented in the manuscript were compared to the expected patterns observed in model organisms and other species. While our data aligns with the anticipated expression patterns for B, C, and Bsister class genes, they deviate for A and E class genes. A possible explanation for these deviations is provided in the Discussion section.

  1. The comparison of A and D homoeolog expression levels across floral organs, mentioned as an aim, is not clearly presented in the results. This analysis should be expanded and highlighted.

=> We expanded the results of homoeolog expression [Lines 428 – 438, 745 – 748]. 

Minor comments

  1. Some figures, particularly Figure 1, could be improved for clarity. Consider using color-coding or different symbols to distinguish between accessions and organs.

=> We revised Figure 1 (now Figure 2) to use the same color-coding for organs as in the expression data.

  1. There is some repetition in the results section, particularly regarding differential expression between organs. This could be condensed for better readability.

=> We tried to revise the results and discussion sections to remove repetition.

  1. Some technical terms (e.g., HEB, DPA) are used without explanation. These should be defined upon first use.

=> We added explanations for both terms: HEB [Lines 379 – 380] and DPA [Lines 127 – 128]

  1. Consider using a table to summarize the key differentially expressed transcription factors in each organ, rather than listing them in the text.

=> In this manuscript, we focused on transcription factors (TFs) that are differentially enriched rather than differentially expressed as TFs with low expression levels can still have significant downstream effects. However, we included STRING analysis results for TFs that are differentially expressed in each floral organ relative to bract, presented as Figure S6.

The manuscript is scientifically sound and makes a significant contribution to understanding floral transcriptomics in cotton. With minor revisions, it is suitable for publication in Plants.

Thank you again for your valuable comments!

Round 2

Reviewer 1 Report

Comments and Suggestions for Authors

The revised manuscript has addressed most of concerns raised and the authors have also improved certain parts. Although I recommended the drawing of a working model to depict the regulation of key genes identified in their work, authors expressed difficulties in adding it. However, the MS can proceed for publication without further comments. 

Reviewer 2 Report

Comments and Suggestions for Authors

The authors have answered my comments and I have no further questions.